# MindSearch: Mimicking Human Minds Elicits Deep AI Searcher

**Zehui Chen**[*1], **Kuikun Liu**[*2], **Qiuchen Wang**[1], **Jiangning Liu**[2], **Wenwei Zhang**[2]
**Kai Chen**[2†], **Feng Zhao**[1†]

[1]MoE Key Laboratory of Brain-inspired Intelligent Perception and Cognition, USTC
[2]Shanghai AI Laboratory
{lovesnow,qiuchenwang}@mail.ustc.edu.cn
{liukuikun,liujiangning,chenkai}@pjlab.org.cn
fzhao956@ustc.edu.cn

## Abstract

Information seeking and integration is a complex cognitive task that consumes enormous time and effort. Search engines reshape the way of seeking information but often fail to align with complex human intentions. Inspired by the remarkable progress of Large Language Models (LLMs), recent works attempt to solve the information-seeking and integration task by combining LLMs and search engines. However, these methods still obtain unsatisfying performance due to three challenges: (1) complex requests often cannot be accurately and completely retrieved by the search engine once; (2) corresponding information to be integrated is spread over multiple web pages along with massive noise; and (3) a large number of web pages with long contents may quickly exceed the maximum context length of LLMs. Inspired by the cognitive process when humans solve these problems, we introduce MindSearch (思·索) to mimic the human minds in web information seeking and integration, which can be instantiated by a simple yet effective LLM-based multi-agent framework consisting of WebPlanner and WebSearcher. The WebPlanner models the human mind of multi-step information seeking as a dynamic graph construction process: it decomposes the user query into atomic sub-questions as nodes in the graph and progressively extends the graph based on the search result from WebSearcher. Tasked with each sub-question, WebSearcher performs hierarchical information retrieval with search engines and collects valuable information for WebPlanner. The multi-agent design of MindSearch enables the whole framework to seek and integrate information parallelly from larger-scale (*e.g.*, more than 300) web pages in **3 minutes**, which is worth **3 hours** of human effort. Based on either GPT-4o or InternLM2.5-7B models, MindSearch demonstrates significant improvement in the response quality in terms of depth and breadth, on both closed-set and open-set QA problems. Besides, responses from MindSearch based on InternLM2.5-7B are preferable by humans to ChatGPT-Web (by GPT-4o) and Perplexity.ai applications, which implies that MindSearch with open-source models can already deliver a competitive solution to the proprietary AI search engine. Code is available at https://github.com/InternLM/MindSearch.

## 1 Introduction

Information seeking and integration is a necessary cognitive process before analysis and decision-making in all walks of life, which usually consumes enormous human efforts and time. The birth of search engines (Brin & Page, 1998; Berkhin, 2005) significantly has reshaped and eased the information-seeking process of human society, however, it still suffers in integrating web information based on complex human intentions. Recently, Large Language Models (LLMs) have showcased remarkable progress in reasoning, language understanding, and information integration across

---

[*]Equal Contribution
[†]Corresponding authors.

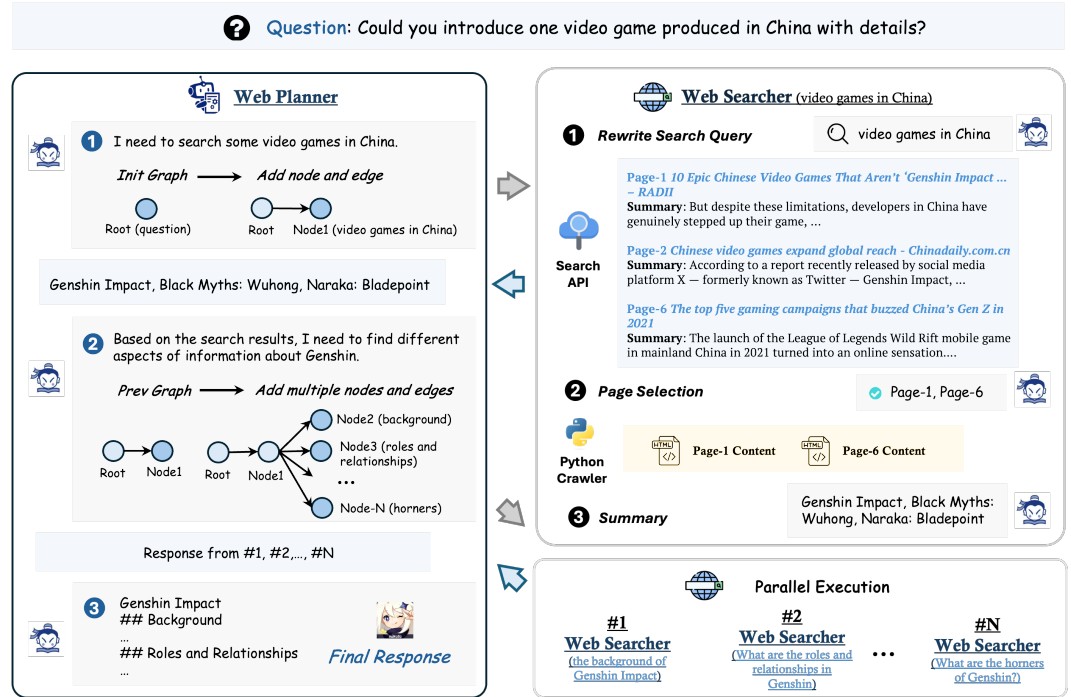

Figure 1: **The overall framework of MindSearch.** It consists of two main ingredients: WebPlanner and WebSearcher. WebPlanner acts as a high-level planner, orchestrating the reasoning steps and multiple WebSearchers. WebSearcher conducts fine-grained web searches and summarizes valuable information back to the planner, formalizing a simple yet effective multi-agent framework.

a variety of domains (Achiam et al., 2023; Team et al., 2024; Touvron et al., 2023; Cai et al., 2024), whereas they struggling to deliver accurate knowledge in responses (Ji et al., 2023; Gu et al., 2024).

The complementary advantages of LLMs and search engines highlights a compelling opportunity for their combination, where the reasoning prowess of LLMs can be complemented by the extensive web information accessible via search engines, potentially revolutionizing the solution of web information seeking and integration. Previous works (Asai et al., 2023; Chan et al., 2024) simply treat the information seeking and integration task as a vanilla retrieve-augmented generation (RAG) task (Chen et al., 2017; Lin et al., 2023). Such a formulation, although straightforward, often results in sub-optimal performance due to a superficial engagement with the depth and complexity of web-based information retrieval, facing three major challenges for more complex user queries:

(1) Real-world problems often require in-depth analysis and proper decomposition of the question before retrieving the related information, which cannot be done by retrieving web pages at once.
(2) The overwhelming volume of searched web pages and massive information noise pose great challenges for LLMs for efficient information integration.
(3) The rapid proliferation of web search content can quickly exceed the maximum context length of LLMs, which further decreases the information integration performance.

Inspired by how human experts solve real-world problems, we propose MindSearch 思·索[1], a simple yet effective LLM-based multi-agent framework, which consists of a WebPlanner (mimic human minds for problem reasoning) and multiple WebSearchers (manage the information seeking process). Given a user query, the WebPlanner first decomposes the query into multiple atomic subquestions that can be parallelly solved and dispatches them to the respective WebSearcher. To further enhance the reasoning ability, WebPlanner models the complex problem-solving process as an iterative graph construction: by predefining a list of standard code interfaces related to the construction of the topological mind graph, WebPlanner is able to progressively decompose the question into sequential/parallel sub-problems by adding nodes/edges in the graph via Python code generation.

---

[1]The Chinese name '思·索' means thinking as human and exploring by searching

Meanwhile, the WebSearcher, tasked with each sub-problem, employs a hierarchical retrieval process to extract valuable data for LLMs, which significantly improves the information aggregation efficiency facing massive search pages. By distributing different aspects of the reasoning and retrieval process to specialized agents, MindSearch effectively reduces the load on each single agent, facilitating a more robust handling of long contexts. It seamlessly bridges the gap between the raw data retrieval capabilities of search engines and the context-understanding power of LLMs.

To validate the effectiveness of MindSearch, we conducted extensive evaluations on both closed-set and open-set question-answering (QA) problems using GPT-4o and InternLM2.5-7B-Chat models. The experimental results demonstrate a substantial improvement in response quality, both in the dimensions of depth and breadth. Moreover, comparative analysis shows that the responses of MindSearch are more preferred by human evaluators over those from existing applications like ChatGPT-Web (based on GPT-4o) and Perplexity Pro. These findings suggest that MindSearch with open-source LLMs can offer a highly competitive solution for AI-driven search engines.

## 2 MindSearch

To effectively synergize the web information retrieval capabilities of search engines and the reasoning and information integration capability of LLMs, MindSearch consists of a WebPlanner and a group of WebSearchers (Fig. 1). WebPlanner first decomposes the user question into sequential or parallel search tasks via reasoning on the graph and determines the next step based on the search feedback (Sec. 2.1). WebSearcher is tasked with the query and performs hierarchical information retrieval on the Internet to answer sub-questions (Sec. 2.2). We also discuss the context management within the scope of the multi-agent design in Sec. 2.3.

### 2.1 WebPlanner: Planning via Graph Construction

The WebPlanner functions as a high-level planner, orchestrating the reasoning steps and coordinating other agents. However, we observed that merely prompting the LLM to plan the entire data workflow architecture does not yield satisfactory performance. Specifically, current LLMs struggle with decomposing complex questions and understanding their topological relationships, leading to coarse-grained search queries. This approach underutilizes the potential of LLMs to serve as intermediaries between humans and search engines, transforming human intentions into step-by-step search tasks and delivering accurate responses.

To enhance the capability of LLM in addressing complex questions, we model the problem-solving process as a directed acyclic graph (DAG). Given a user question $Q$, the solution trajectory is represented as $G(Q) = \langle V, E \rangle$, where $V$ is a set of nodes $v$, each representing an independent web search, including an auxiliary START node (the initial question) and an END node (the final answer). $E$ represents directed edges indicating the reasoning topological relationships between nodes (search contents). This DAG formalism captures the complexity of finding the optimal execution path, providing a more formal and intuitive representation for LLMs.

Leveraging the superior performance of current LLMs on code tasks (Guo et al., 2024; Roziere et al., 2023), we explicitly prompt the model to interact with the graph through code writing. To achieve this, we predefined atomic code functions to add nodes or edges to the graph (Step 1 and 2 in Figure 2). At each turn, the LLM first reads the entire dialogue, including previously generated code and web search results, then outputs thoughts and new code for reasoning on the mind graph, which is executed with a Python interpreter. During execution, once a node is added to the reasoning graph, it invokes a WebSearcher to execute the search process and summarize the information. Since the newly added nodes are only dependent on nodes generated in previous steps, we can parallel them to achieve a much faster information aggregation speed. When all information is collected, the planner produces the final response by adding the end node (Step 3 in Figure 2).

By integrating with the Python interpreter, WebPlanner interacts with the graph through unified code actions, dynamically constructing the reasoning path. This "code as planning" process enables the LLM to fully leverage its superior code generation ability, benefiting control and data flow in long-context scenarios and leading to better performance in solving complex problems.

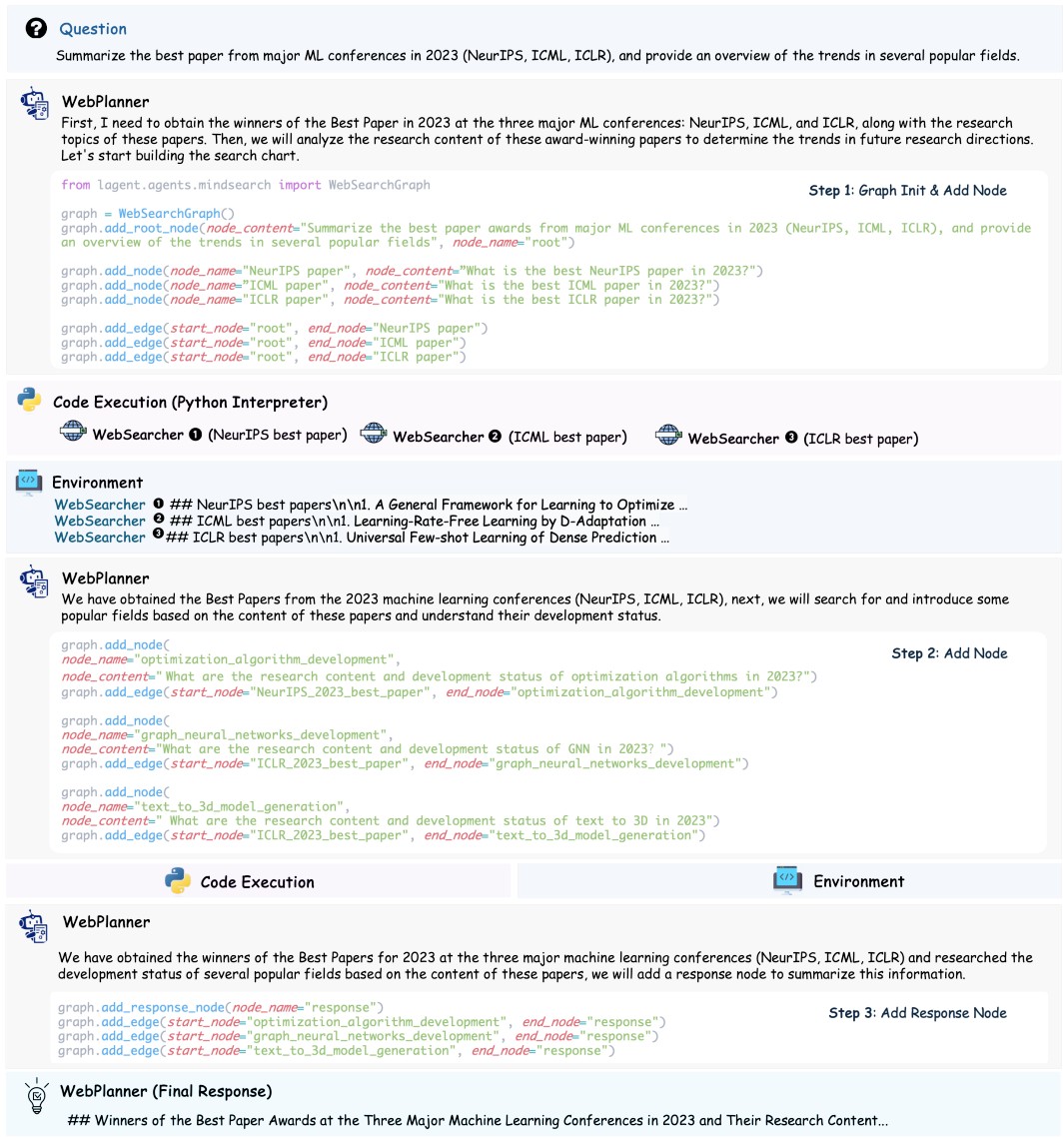

Figure 2: **A concrete example of how WebPlanner addresses the question step by step via planning as coding.** During each turn, WebPlanner outputs a series of thoughts along with the generated code. The code will be executed and yield the search results to the planner. At the last turn, the WebPlanner directly provides the final response without any code generation.

## 2.2 WebSearcher: Web Browsing with Hierarchical Retrieval

WebSearcher acts as a sophisticated RAG (Retrieve-and-Generate) agent with internet access, summarizing valuable responses based on search results (Figure 3). Due to the massive content available on the web, it is challenging for LLMs to process all related pages within a limited context length (*e.g.* 8K tokens). To address this, we employ a straightforward coarse-to-fine selection strategy. Initially, the LLM generates several similar queries based on the assigned questions from the Web-Planner to broaden the search content and thus improve the recall of relevant information. These queries are then executed through various search APIs, such as Google, Bing, and DuckDuckGo, which return key contents including web URLs, titles, and summaries. The search results are automatically merged based on the web URLs, and the LLM is prompted to select the most valuable pages for detailed reading. The full content of the selected web URLs is then added to the input of LLM. After reading these results, the LLM generates a response to answer the original question

based on the search results. This hierarchical retrieval approach significantly reduces the difficulty of navigating massive web pages and allows to efficiently extract highly relevant information with in-depth details.

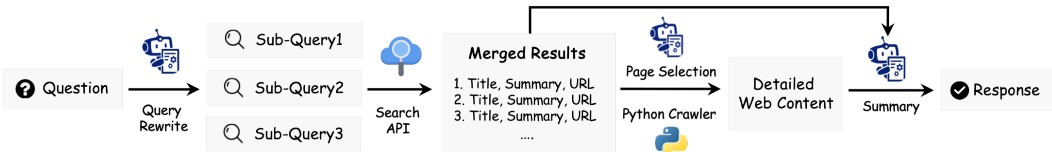

Figure 3: **A detailed working pipeline of WebSearcher.** It comprises at most 4 steps: query rewrite, search content aggregation, detailed page selection, and final summarization.

## 2.3 LLM CONTEXT MANAGEMENT IN MINDSEARCH

MindSearch provides a simple multi-agent solution to complex information seeking and integration with search engines. Such a paradigm also naturally enables long-context management among different agents, which improves the overall efficiency of the framework, especially under circumstances that require the model to quickly read plenty of web pages. Since the WebPlanner distributes the search tasks into separate search agents and only relies on the searched results from WebSearcher, WebPlanner can purely focus on the decomposition and analysis of the user question without being distracted by the over-length web search results. Meanwhile, each WebSearcher only needs to search contents for its tasked sub-query, without distraction from other contents. Thanks to the explicit role distribution, MindSearch greatly reduces context computation during the whole process, delivering an efficient context management solution to long-context tasks for LLM. Such a multi-agent framework also provides a straightforward and simple long-context task construction pipeline for training single LLMs, which is also observed in (Team, 2024). Eventually, MindSearch collects and integrates related information from more than 300 pages in less than 3 minute, which could take human experts about 3 hours to finish a similar cognitive workload.

Due to the explicit context state transfer across multiple agents, we need to carefully handle the context during the whole workflow. We empirically find simply focusing the decomposed query from the Planner may lose useful information during the information collection phase due to the local receptive field inside the search agent. How to effectively handle the context between multiple agents is non-trivial. We find that the constructed topological relations through the directed graph edges help us easily handle the context across different agents. More specifically, we simply prefix the response from its father node as well as the root node when executing each search agent. Therefore, each WebSearcher can effectively focus on its sub-task without losing the previous related context as well as the final goal.

## 3 EXPERIMENTS

We evaluate MindSearch on two primary categories of Question Answering (QA) tasks: closed-set QA and open-set QA, which reflects both the subjective and objective judgment of MindSearch. For a fair comparison, all models only have access to the Internet through BING search API, and no extra reference sources are considered.

### 3.1 OPEN-SET QA

#### 3.1.1 IMPLEMENTATION DETAILS

To better gauge the utility and search performance, we carefully curate 100 real-world human queries and collect responses from MindSearch (InternLM2.5-7b-chat (Cai et al., 2024)), Perplexity.ai (its Pro version), and ChatGPT with search plugin (Achiam et al., 2023). We ask five human experts to manually select their preferred responses, in terms of the following three aspects:

- **Depth**: Depth refers to the thoroughness and profundity of an answer. A response with depth provides detailed information and delves into the intricacies of a question.

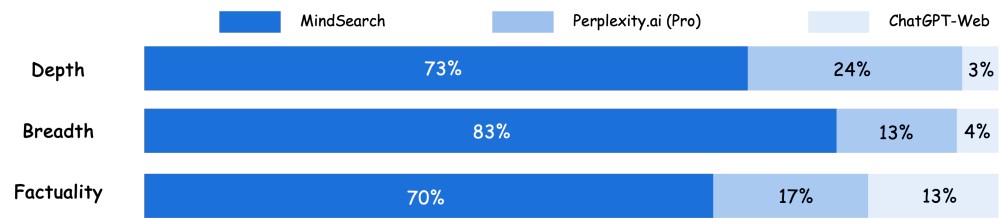

Figure 4: **Subjective evaluation results judged by human experts on open-set QA questions.** MindSearch outperforms ChatGPT-Web and Perplexity.ai Pro by a large margin in terms of depth, breadth, and facticity.

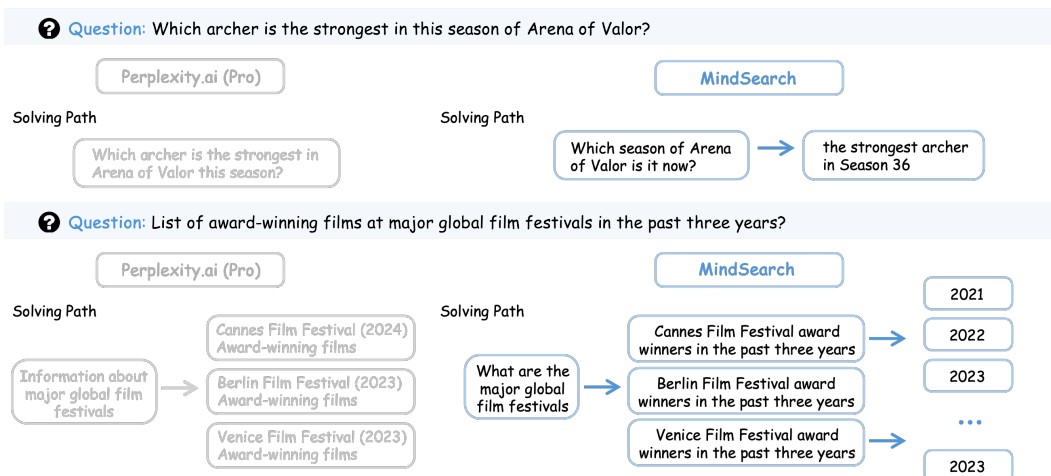

Figure 5: **Solution trajectory comparison between MindSearch and Perplexity.ai (Pro) on the same question.** MindSearch provides more detailed and proper responses thanks to its fine-grained searches.

- **Breadth**: Breadth pertains to the scope and diversity covered by an answer. A response with breadth touches on various aspects of the question or multiple related fields, offering different perspectives or solutions.

- **Factuality**: Factuality is the degree to which an answer is accurate and fact-based. It should be grounded in reliable data and information, avoiding errors or misleading content, and ensuring the truthfulness and credibility of the information provided.

The final results are determined based on major votes. During the evaluation, the correspondence between the response and its method is invisible to the evaluators to guarantee fairness.

### 3.1.2 RESULTS AND ANALYSIS

The evaluation results are depicted in Figure 4 and we also provide quantitative results in Figure 5. From Figure 4, we can observe an absolute improvement in terms of the depth and breadth of the model response, which validates the superiority of our proposed WebPlanner. By integrating code into the DAG construction phase, LLM is able to progressively decompose the complex problem into executable queries while balancing the tradeoff between time efficiency and the exploration of the search space. Besides, MindSearch goes through more fine-grained search topics about the question, therefore providing more compact and detailed responses compared to other models. However, MindSearch does not yield much better performance in terms of facticity, compared to breadth (70% vs 83%). We suspect that more detailed search results may distract the concentration of the model on the initial problem, especially when LLM holds incomplete long-context capability. Therefore, a natural future work of MindSearch is to alleviate the hallucination issues during the web browsing process.

Table 1: **Performance comparison on various closed-set QA tasks**. We select two representative LLMs: GPT-4o (close-sourced) and InternLM2.5-7b-chat (open-sourced).

| Model | Bamboogle | Musique | | | HotpotQA | | | AVG |
| --- | --- | --- | --- | --- | --- | --- | --- | --- |
| | | 2-hop | 3-hop | 4-hop | Easy | Medium | Hard | |
| Closed-Souced LLM (GPT-4o) | | | | | | | | |
| w/o Search Engine | 70.4 | 54.0 | 22.0 | **20.0** | 73.0 | 69.0 | 66.0 | 53.5 |
| ReAct Search | 75.2 | 48.0 | 25.0 | 13.3 | **81.0** | 73.0 | 70.0 | 55.1 |
| MindSearch | **76.8** | **60.0** | **35.0** | 14.6 | 80.0 | **74.0** | **78.0** | **59.8** |
| Open-Sourced LLM (InternLM2.5-7b-chat) | | | | | | | | |
| w/o Search Engine | 34.0 | 28.0 | 10.0 | 17.3 | 47.0 | 26.0 | 40.0 | 28.9 |
| ReAct Search | 55.2 | 38.0 | 17.0 | 16.0 | 69.0 | 56.0 | 49.0 | 42.9 |
| MindSearch | **67.8** | **46.0** | **20.0** | **18.6** | **69.0** | **66.0** | **57.0** | **49.2** |

In addition to quantitative results, we also provide a qualitative response comparison between Perplexity.ai (Pro) and MindSearch to deliver an intuitive understanding of their performance. From Figure 5, we can observe that MindSearch yields more concrete and detailed responses. We empirically find that our better responses can be attributed to the proper planning search paths compared to Perplexity.ai, which also indicates that how to decompose the human intention is the key step to the final problem.

## 3.2 CLOSED-SET QA

### 3.2.1 IMPLEMENTATION DETAILS

We extensively evaluate our approach on a wide range of closed-set QA tasks, including Bamboogle (Press et al., 2022), Musique (Trivedi et al., 2022), and HotpotQA (Yang et al., 2018). To further validate the generalization of our approach, we select both closed-source LLM (GPT-4o) and open-source LLM (InternLM2.5-7b-chat) as our LLM backend. Since our approach adopts a zero-shot experimental setting, we utilize a subjective LLM evaluator (GPT4-o) to gauge the correctness of HotpotQA.

### 3.2.2 RESULTS AND ANALYSIS

In Table 1, we compare our approach with two straight-forward baselines: raw LLM without search engines (w/o Search Engine), and simply treating search engines as an external tool and adopting a ReAct-style interaction (ReAct Search). We can conclude that MindSearch significantly outperforms its vanilla baselines by a large margin (4.7% on GPT-4o and 6.3% on InternLM2-7b), validating the effectiveness of the proposed method. These advantages are amplified when transferring from closed-sourced LLMs to open-sourced LLMs, which further proves that MindSearch provides a simple approach to enhance weak LLMs with broader knowledge and alleviate hallucination issues. When taking a closer look at the difficulty level of HotpotQA, we observe that most improvements are derived from the hard set, which also indicates that MindSearch are more adept at solving complex questions compared to other approaches.

## 3.3 ABLATION STUDIES

In this section, we conduct detailed ablation studies aiming to gain a deeper understanding of our approach. Without loss of generality, all experiments are conducted with InternLM2.5-7b-chat on HotpotQA if not specified.

### 3.3.1 ABLATIONS ON WEBPLANNER

To validate the efficiency of our proposed WebPlanner, we compare our graph-based code planning strategy with its two vanilla baselines, namely ReAct (Yao et al., 2022b), and CodeAct (Wang et al., 2024). For ReAct, we prompt the WebPlanner to invoke WebSearcher with a classical predefined Re-

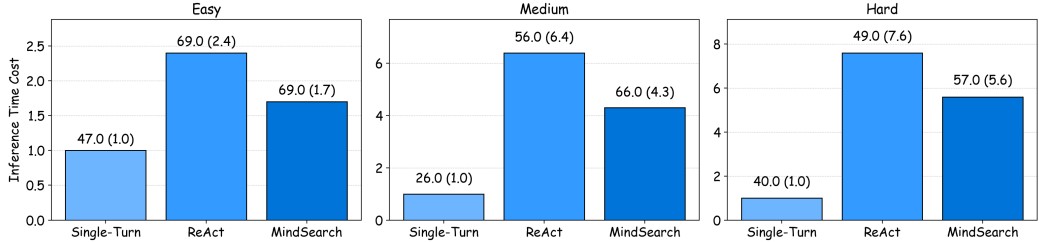

Figure 6: **Inference cost vs. search performance among various inference strategies on HotpotQA.** The scores above each bar are search performance and inference cost, respectively. We normalize the inference time cost by single-turn.

Table 2: Ablations on the planning strategies adopted in WebPlanner. For a fair comparison, we keep WebSearcher the same for all entries. '-' denotes no access to web search during inference time.

| Strategy | Performance |
|---|---|
| - | 37.6 |
| ReAct | 58.0 |
| CodeAct | 61.3 |
| Ours | **64.0** |

Table 3: Ablations on the components adopted in WebSearcher. 'MQG' denotes multi-query generation and aggregation, 'PPC' denotes prefix previous context, and 'PS' denotes the page selection process.

| Method | Performance |
|---|---|
| Ours | **64.0** |
| w/o MQG | 60.6 |
| w/o PPC | 63.3 |
| w/o PS | 58.0 |

Act JSON format, *i.e.,* `Thought: str; Action: WebSearcher(List[str])` at each step. As for CodeAct, we wrap WebSearcher as a function and prompt the WebPlanner to use it by writing code. For a fair comparison, we keep the WebSearcher the same for all entries. The results are shown in Table 2. We can observe that integrating code generation provides unified action space and therefore yields better performance compared to ReAct, which proves the correctness of our adoption of code as planning. Then, when converting the original function calling process into the graph construction, the performance continues to improve, especially on the hard set of HotpotQA, which further validates the superiority of our proposed planning as graph construction.

### 3.3.2 ABLATIONS ON WEBSEARCHER

The search quality of WebSearcher determines the upper bound of the final response from WebPlanner. Therefore, how to efficiently aggregate highly related web pages via search engines and provide valuable summary information is of great importance. In this part, we independently remove the multi-query generation and aggregation, prefix previous context, and page selection processes, and report the results in Table 3. Still, we keep the WebPlanner the same for all entries for a fair comparison. As can be seen, each module contributes to the final performance and multi-query generation has the most influence.

### 3.4 INFERENCE TIME SCALING VS. SEARCH PERFORMANCE

Properly scaling test-time computation enables LLMs to improve their outputs by a considerable margin (Snell et al., 2024; Kumar et al., 2024). OpenAI o1 (OpenAI, 2024) has recently shown that a detailed chain-of-thought can dramatically enhance the reasoning ability of language models on various downstream tasks. In this section, we discuss the relationship between inference time scaling and search performance in MindSearch, which systematically analyzes such a trade-off under the AI search engine domain. Specifically, we compare three types of search patterns in Figure 6: (1) single-turn search, with only one search call for each question (adopted by most current AI search engines), (2) multi-turn search with ReAct, and (3) multi-turn search with MindSearch. The last two patterns allow LLMs to scale up their inference computation with multiple web search interactions (during experiments, we limit the max interaction turn to 10 since we observe limited performance

gains by enlarging this hyperparameter). Single-turn search achieves a passing grade on the easy level and uses the shortest time, which actually meets the need for a large portion of real-world usage (possibly the reason why most AI search engines adopt this pattern). By enabling linear-scale reasoning through ReAct, we are able to observe that the performance improves at the cost of more inference costs. Compared to ReAct, MindSearch gets more efficient scaling performance with less inference steps, which indicates that MindSearch provides a better scaling strategy for improving the search performance.

## 4 RELATED WORK

### 4.1 TOOL UTILIZATION WITH LLM

The Tool Learning framework empowers LLMs to seamlessly integrate with a variety of tools (Qin et al., 2023; Hao et al., 2024; Zhuang et al., 2024; Chen et al., 2023), such as search engines (Chan et al., 2024), databases (Parisi et al., 2022), and APIs (Li et al., 2023; Patil et al., 2023), offering dynamic solutions to complex problems. This integration is not only beneficial for enhancing the interpretability and trustworthiness of LLMs but also for improving their robustness and adaptability across diverse tasks, including reducing hallucinations (Ji et al., 2024), code generation (Gou et al., 2023), and question answering (Chen et al., 2024). Recent research has focused on enhancing the tool integration component of Tool Learning systems. Works such as (Huang et al., 2023; Shen et al., 2023; Schick et al., 2024) have concentrated on improving the retrieval mechanisms, ensuring that LLMs can access the most pertinent tools for a given task. Other studies, like (Qian et al., 2023; Yuan et al., 2023), aim at refining the LLMs' ability to effectively utilize the retrieved information, optimizing the reading and comprehension processes within the framework.

### 4.2 RAG WITH LLM

RAG demonstrates significant advantages in addressing knowledge-intensive problems, especially in open-domain scenarios with the integration of search engines (Chen et al., 2017; Li et al., 2017). RAG allows LLMs to integrate with the retriever, providing timely information and offering effective solutions. Moreover, RAG is also applied in various tasks such as reducing hallucinations (Shuster et al., 2021; Gu et al., 2024), code generation (Zhou et al., 2022), and question answering (Lewis et al., 2020). Recently, some work (Karpukhin et al., 2020; Xiong et al., 2020; Qu et al., 2020) focuses on enhancing the retrieval component of RAG systems, while others (Izacard & Grave, 2020; Borgeaud et al., 2022; Yu et al., 2021; Lei et al., 2017) enhances the language model's ability as a reader to optimize the framework.

With the advancement of LLM capabilities, some researchers have begun to reoptimize frameworks and redesign methodologies for model training (Luo et al., 2023a; Qiao et al., 2024). SAIL (Luo et al., 2023b) trains LLM to be more focused on credible and informative search results. Self-RAG (Asai et al., 2023) enables LMMs to independently fetch, introspect, and augment their text generation capabilities. RQ-RAG (Chan et al., 2024) enhances query formulation by learning to refine queries through an iterative process. Searchain (Xu et al., 2024) introduces chain-of-query (CoQ) to iteratively refine the reasoning of graph to resolve complex problems. Our work integrates web search capabilities into LLMs, enhancing response quality by retrieving valuable information from the Internet.

### 4.3 WEB AGENTS

Web automation agents have evolved from question-answering tools to sophisticated systems capable of complex web interactions. Early models like WebGPT (Nakano et al., 2021) and WebGLM (Liu et al., 2023) primarily addressed QA tasks, while recent advancements have shifted towards more dynamic operations (Yao et al., 2022a; He et al., 2024). MindAct (Deng et al., 2024), WebAgent (Gur et al., 2023), SeeAct (Zheng et al., 2024), and SeePlanAct (Yoran et al., 2024) represent this progression, with the latter showing exceptional web navigation despite deployment challenges due to its size. AutoWebGLM (Lai et al., 2024) offers a practical alternative with robust capabilities and a more compact model size. The incorporation of reinforcement learning (Bai et al., 2024) and behavior cloning techniques (Zheng et al., 2024; Patel et al., 2024) paves the way for even more

autonomous and efficient web automation, moving the field towards scalable and versatile solutions for real-world applications. This paper mainly focuses more on the web information-seeking and integration task with search engines instead of web browsing, and solves the main challenges with a multi-agent framework.

## 5 Conclusion and Future Work

This paper introduces MindSearch, a novel LLM-based multi-agent framework for complex web information-seeking and integration tasks, by more comprehensively leveraging the strengths of both search engines and LLMs. MindSearch conducts effective and sufficient decomposition of complex queries followed by hierarchical information retrieval to improve the precision and recall of the retrieved relevant web information, by modeling the problem-solving process as an iterative graph construction. The multi-agent design distributes the cognitive load among specialized agents, facilitating robust handling of complex and lengthy contexts. Extensive evaluations on closed-set and open-set QA problems using GPT-4o and InternLM2.5-7B models demonstrated significant advantages in the response quality of MindSearch. The results that human evaluators preferred the responses from MindSearch over those from ChatGPT-Web and Perplexity.ai indicate its competitive edge in AI-driven search solutions. However, there exist some limitations in this work: the citation quality of the web search references is not evaluated comprehensively, considering the extremely diverse and subjective evaluation of AI web search engines compared to closed-set QA reference evaluation. Besides, MindSearch does not support visual inputs, and cannot interact with web pages, which is a promising and more complex scenario in real-world applications. We leave them for future work and will continue to explore them in MindSearch. We wish this work pave the way for future research on multi-agent framework for solving human-level complex cognitive tasks.

## 6 Acknowledgement

This work was supported by the Anhui Provincial Natural Science Foundation under Grant 2108085UD12. We acknowledge the support of GPU cluster built by MCC Lab of Information Science and Technology Institution, USTC.
We would like to express our sincere gratitude to Jiaye Ge for her outstanding technical insights, product design skills, and coordination abilities. We also thank Zhongying Tu, Ying Zhao, Fang Fang, and Yiting Wang for their efficient execution in developing the project demo. Our gratitude extends to Xingyuan Liu, Shuaike Li, Zike Pan, Weijia Song, and Yuzhe Gu for their efforts in project suggestion.

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

## A    COMPARISON WITH OTHER STATE-OF-THE-ARTS

In this section, we compare our approach with more competitive web search counterparts, including self-ask (Press et al., 2022), CodeAct (Wang et al., 2024), and Searchain (Xu et al., 2024). All experiments are conducted on the HotpotQA dataset with InternLM-7b-2.5 and the results are shown in Table 4. We can find that MindSearch consistently outperforms other state-of-the-art approaches by a large margin. Specifically, Searchain shares a similar spirit to our approach which formulates the reasoning as a query-of-chain. However, at each revised step, Searchain needs to re-generate the whole reasoning chain due to its weakness in long-context reasoning, which is time-consuming and fallible. Thanks to the multi-agent design, MindSearch is able to reason at each step immediately when necessary.

Table 4: Comparison with state-of-the-art approaches for web search tasks on HotpotQA dataset.

| Method | Easy | Medium | Hard | Average |
|---|---|---|---|---|
| ReAct (Yao et al., 2022b) | 69.0 | 56.0 | 49.0 | 58.0 |
| self-ask (Press et al., 2022) | 67.0 | 59.0 | 49.0 | 58.3 |
| CodeAct (Wang et al., 2024) | 70.0 | 63.0 | 51.0 | 61.3 |
| Searchain (Xu et al., 2024) | **70.0** | 61.0 | 54.0 | 61.6 |
| MindSearch (Ours) | 69.0 | **66.0** | **57.0** | **64.0** |

## B    TIME EFFICIENCY ON MINDSEARCH VS. HUMAN LABELERS

In this section, we provide detailed time cost measurement in terms of the human labelers and MindSearch on 10 complex research questions. We randomly distributed 10 questions to 5 human experts and asked them to collect information with web search engines. After retrieving enough data, each labeler is requested to write a detailed answer to the question. Human labelers spend 19h17min to accomplish the search tasks while MindSearch only takes 23 min. It can be seen that there exists a relationship between MindSearch and human labelers with 1 min vs 1-hour time efficiency. Furthermore, we also analyze the time cost of one human labeler when labeling one question. 47 minutes are taken to collect Information by searching and reading multiple (100+) web pages and another 74 minutes are required to write a detailed response (about 3,000 words).

## C    MORE ANALYSIS ON WEBPLANNER

### C.1    NUMBER OF HOPS VS DEPTH OF DAG

We conduct experiments on Musique with GPT-4o with 2, 3, 4 hops to study the relationships between the number of hops and the depth of the DAG. From Table 5, it can be seen that the depth of the tree increases with the number of hops monotonically, which fits our expectations. However, the number of hops is not identical to the depth of the tree, for example when the number of hops is 3, the depth of the tree is 1.2. There are two reasons: (1) MindSearch allows parallel execution, which only increases the tree by one but may resolve multiple questions at the current step, and (2) despite the question claiming 2 or more hops, there exist short-cuts or simplifications in the question, resulting shorter search path.

Table 5: Experimental results on the number of hops of questions vs. the depth of the generated DAG by WebPlanner on Musique dataset.

| Num Hops | Depth |
|---|---|
| 2 | 1.1 |
| 3 | 1.2 |
| 4 | 1.6 |

## C.2 COST ANALYSIS ON THE NUMBER OF SEARCH QUERIES

We analyze the number of queries generated by MindSearch and ReAct. Surprisingly, MindSearch generates 0.3 fewer queries on average compared to ReAct (3.2 vs 3.5). We find that due to the weaker ability of ReAct to decompose the question, ReAct actually spends more queries repeatedly searching for some keywords, which is useless and inefficient. However, MindSearch can effectively utilize the reasoning ability of LLM to search for more accurate queries, therefore, yielding fewer search times for each problem.

## D GENERALIZATION TO OTHER LANGUAGE MODELS

In this section, we experiment with MindSearch on various accessible large language models to further validate the generalization of our approach and the extensibility of the DAG-based code reasoning interface. We select three representative models: DeepSeekv2 (Liu et al., 2024), Qwen-2.5-7b (Yang et al., 2024), and GLM-4-9b (GLM et al., 2024), and the results are shown in Table 6. It can be seen that MindSearch can easily adapt to various models with little adaptation, which further proves the effectiveness of our approach.

Table 6: Experimental results on various LLMs for web search tasks on HotpotQA dataset with MindSearch.

| Model | Easy | Medium | Hard | Average |
|---|---|---|---|---|
| DeepSeek-V2 (Liu et al., 2024) | 70.0 | 71.0 | 68.0 | 69.6 |
| Qwen-2.5-7b (Yang et al., 2024) | 62.0 | 59.0 | 52.0 | 57.6 |
| GLM4-9b (GLM et al., 2024) | 65.0 | 60.0 | 55.0 | 60.0 |

## E ANALYSIS ON ERROR CORRECTION IN MINDSEARCH

In this section, we comprehensively demonstrate the error correction ability of MindSearch, which reflects the superiority of the code interface for planning, and the effectiveness of DAG protocol. We delicately select several typical examples of how MindSearch recovers from its previous mistakes, which helps the readers gain an intuitive understanding of how MindSearch deals with occasional search accidents.

### E.1 ERROR CORRECTION WITH CODE EXECUTION FEEDBACK

MindSearch adopts the code as the execution protocol during the agent running, which has two advantages: (1) large language models are more skillful in generating structure language, compared to natural form, and (2) code protocol provides us with validation mechanism: when WebPlanner generates wrong code or invalid planning sentence, we can simply validate it in the Python code interpreter and seed the exception back to the LLM for re-generation. It is inevitable to generate the wrong token during the text generation phase, due to the sampling techniques in seq2seq models. Therefore, it is necessary to maintain a validatable protocol for self-correction. Figure 7 demonstrates how MindSearch observes the mistakes during the node execution phase and corrects itself with the help of the error messages provided by the code executor.

### E.2 ERROR CORRECTION WHEN WEBSEARCHER CANNOT FOUND RELATED INFORMATION

MindSearch introduces a multi-agent framework to address the extremely long context encountered in web search scenarios. However, more communication among agents indicates that MindSearch needs to handle various responses from multiple WebSearchers. In order to avoid unexpected responses from WebSearchers, we limit the response content of each WebSearcher to either related information about the topic or information not found. In Figure 8, we demonstrate that MindSearch is able to successfully recover from previous search results and regenerate the new search queries with "more information about the final match of UEFA Euro 2020", which finally find the corner

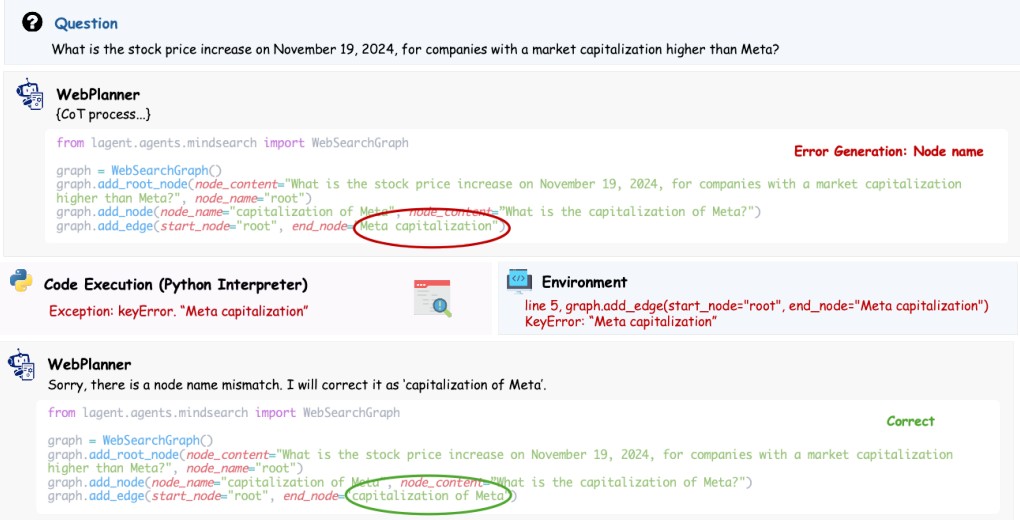

Figure 7: Illustration of error self-correction by MindSearch with the help of the error messages provided by the code executor: MindSearch regenerates the code which correctly fixes the node name error in the next turn.

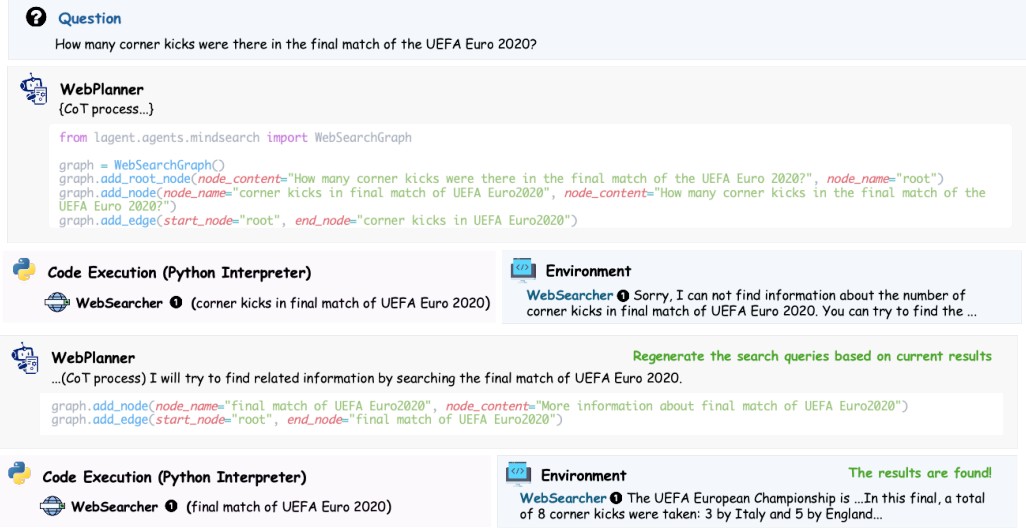

Figure 8: Illustration of error self-correction by MindSearch when WebSearcher responds with 'not founded' feedback. MindSearch regenerates the query and successfully retrieves the expected information during the next search.

kicks in the Wikipedia pages in WebSearcher. Besides, there are also some cases that cannot be found without web interaction. MindSearch enables the model to directly generate the response node when several attempts fail and directly give up the answer, which avoids the model falling into a repeatedly and meaningless loop (Figure 9).

## F    DETAILS ON OPEN-SET EVALUATION PHASE

In this section, we provide more details on the open-set evaluation process for better reproducibility.

- The labelers have no prior knowledge and information about the responses generated by the model.

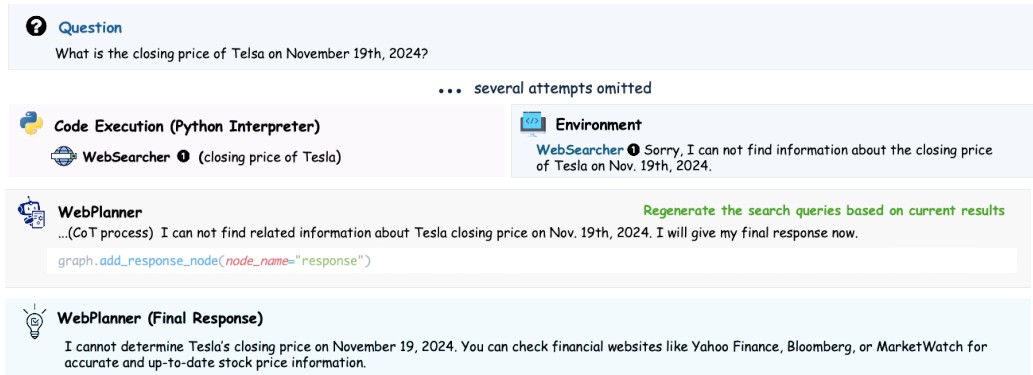

Figure 9: Illustration of the self-conclusion of MindSearch when the information cannot be found within the current search engine. It guarantees that MindSearch will not get stuck in certain tasks and can finally give the answer.

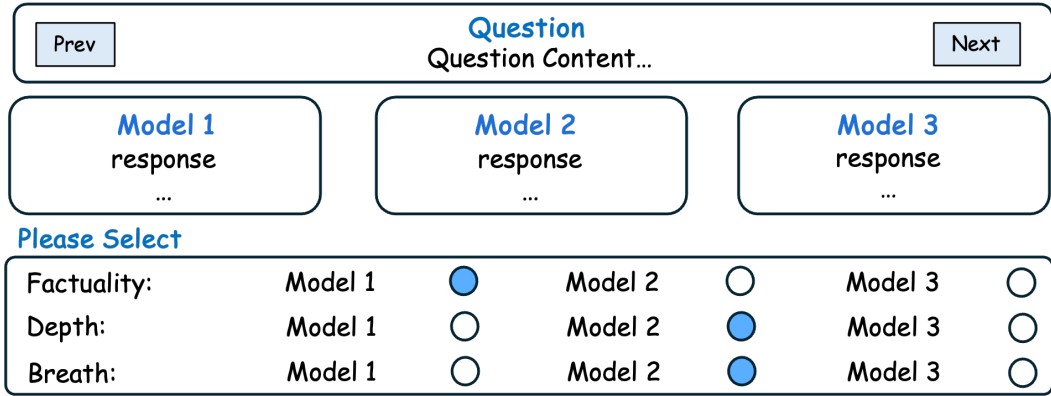

Figure 10: **Open-set evaluation labeling interface illustration.** We do not expose the source of the response to the labeler by replacing them with an anonymous model number, by randomly sampling from 1,2,3 for each question.

- When we ask the labeler to choose the best response, we remove the source method by replacing it with "model1", "model2", and "model3". In order to avoid the labeler making any assumption of the sequence of models 1,2,3, we randomly shuffle the order of the responses before presenting them to the labeler.

- The labelers will be asked to choose the best response from the three responses generated by the model according to three aspects: factuality, depth, and breath.

In order to gain a straightforward understanding of our evaluation process, we provide an illustration of our labeling interface (see Figure 10), where human experts can select the best response based on the factuality, depth, and breath with a simple choice. Besides, we allow labelers to move back to the previous question to re-check/correct their original choices. Apart from the interface demonstration, we also provide a response sample from MindSearch, Perpexity.ai Pro, and ChatGPT-Web in the following:

***Question:*** *Recently, Chang'e-6 returned samples from the far side of the Moon. First, analyze the technical challenges of sampling and returning from the lunar far side, then detail how each challenge was overcome. Compare this achievement with the Apollo 11 lunar landing mission, assessing China's contribution to international space exploration through this successful lunar mission.*

**Response from MindSearch**

The recent success of China's Chang'e-6 mission in returning samples from the far side of the Moon marks a significant milestone in human space exploration. This achievement not only showcases China's technological advancements but also highlights its contributions to international space exploration. Below, we delve into the technical challenges faced during the mission, how these challenges were overcome, and compare this achievement with the Apollo 11 lunar landing mission.

### Technical Challenges and Overcoming Them

#### 1. Communication Difficulties

The lunar far side is permanently shielded from Earth's view due to the Moon's synchronous rotation. This means that direct communication between the lunar far side and Earth is impossible. To address this, China deployed the Queqiao 2 relay satellite in lunar orbit. This satellite acts as an intermediary, enabling data transmission and control signals to be sent to the Chang'e 6 probe on the lunar far side [[6]][[2]].

#### 2. Navigation Challenges

Navigating to the lunar far side poses significant difficulties due to its lack of direct Earth-based communication. This necessitates the use of autonomous navigation systems. The Chang'e 6 mission employs a combination of orbiter, lander, ascender, and reentry capsule components to perform its mission. The lander-ascender combination will make a soft landing in the South Pole-Aitken Basin, the largest and oldest crater on the lunar far side. The navigation system uses relative measurements between a halo orbiter and surface assets to achieve accuracies of tens of meters on the lunar far side surface [[1]][[7]].

#### 3. Power Supply and Energy Management

Power availability is a critical enabler for lunar activities. The Chang'e 6 mission relies on a combination of solar power and onboard energy storage solutions. The LunaGrid service, which consists of power-generating stations and power transmission cables, ensures continuous power supply to lunar surface assets for multi-year operations [[9]][[8]].

#### 4. Sample Collection and Return

Collecting samples from the lunar far side involves advanced robotic systems and mechanisms. The Chang'e 6 probe is equipped with a drill and a mechanical arm to gather surface and underground samples. The collected samples are vacuum-sealed in a metal container and transferred to the reentry capsule for return to Earth. The entire process is designed to be efficient and automated, minimizing the risk of sample contamination and ensuring the integrity of the samples [[6]][[3]].

#### 5. Scientific and Technical Advancements

The lunar far side offers unique scientific opportunities due to its distinct geological features and electromagnetic environment. The samples collected by Chang'e 6 are expected to provide valuable insights into the Moon's early evolution, volcanic activities, impact history, and the record of galactic activity [[4]][[5]].

### Comparison with Apollo 11

The Apollo 11 mission, which successfully landed astronauts Neil Armstrong and Buzz Aldrin on the Moon in 1969, remains one of the most iconic achievements in human history. While Apollo 11 focused on landing humans on the Moon and conducting scientific experiments, Chang'e-6's primary objective was to collect samples from the Moon's far side

and return them to Earth. Both missions represent significant advancements in space exploration, but they differ in their goals and methodologies.

#### Technical Challenges

Both Chang'e-6 and Apollo 11 faced significant technical challenges in their missions. The Apollo 11 mission required precise calculations and engineering to ensure a safe landing on the Moon's surface, while Chang'e-6 had to navigate the lunar far side, which is not visible from Earth due to the Moon's synchronous rotation. The far side of the Moon presents additional challenges, including the lack of communication with Earth during the mission's critical phases.

To overcome these challenges, Chang'e-6 utilized a relay satellite, Queqiao-2, to maintain communication with Earth during the mission's critical phases. This satellite played a crucial role in ensuring the successful return of the samples to Earth. Similarly, Apollo 11 relied on extensive ground-based tracking and communication systems to monitor the spacecraft's progress and ensure a safe return.

### China's Contribution to International Space Exploration

China's involvement in the Chang'e-6 mission highlights its growing role in international space exploration. The mission carried four international payloads, including one developed by Pakistan, which marked Pakistan's first venture into lunar exploration. This collaboration demonstrates China's commitment to fostering international cooperation in space science and technology.

Furthermore, China's lunar exploration program, including the Chang'e series of missions, has provided a platform for international scientific research. The samples collected by Chang'e-6 are expected to contribute significantly to the global understanding of lunar geology and the Moon's formation and evolution. This collaborative approach underscores China's role as a key player in the global space community.

### Conclusion

The success of Chang'e-6 in collecting samples from the Moon's far side and returning them to Earth represents a significant achievement in human space exploration. While the Apollo 11 mission remains a landmark in human history for its successful lunar landing, Chang'e-6's focus on sample collection from the far side of the Moon showcases China's technological prowess and commitment to international cooperation in space exploration. Through these missions, China has made substantial contributions to the global understanding of the Moon and the broader field of space science.

---

**Response from Perplexity.ai**

The recent success of the Chang'e-6 mission in returning samples from the far side of the Moon marks a significant achievement in lunar exploration. This endeavor presented several technical challenges due to the unique conditions and requirements of sampling and returning from this less-explored region.

## Technical Challenges of Sampling and Returning from the Lunar Far Side

1. **Communication Limitations**: The far side of the Moon is always facing away from Earth, making direct communication impossible. This necessitated the use of relay satellites to maintain contact.
2. **Landing Precision**: Achieving a precise landing on the far side, where terrain is rugged and less understood, posed significant risks for the lander.

3. **Sample Collection and Preservation**: Collecting samples without contamination and ensuring their integrity during transit back to Earth was crucial.
4. **Thermal Control**: The lunar environment experiences extreme temperature fluctuations, which could affect both the lander and the samples.
5. **Power Supply**: The mission required a reliable power source for operations, especially during the long lunar night.

## Overcoming the Challenges

1. **Communication Limitations**:
To address communication issues, Chang'e-6 utilized the Tianhe relay satellite, which orbited the Moon and facilitated continuous communication between the lander and mission control on Earth[1].
2. **Landing Precision**:
Advanced navigation systems were employed, including high-resolution terrain mapping and autonomous landing technology, to ensure accurate touchdown on the targeted site[1].
3. **Sample Collection and Preservation**:
The mission utilized a specialized robotic arm designed for lunar soil collection. This arm was equipped with tools to minimize contamination and preserve sample integrity during collection and storage[1].
4. **Thermal Control**:
Chang'e-6 was equipped with thermal insulation materials and active heating systems to protect sensitive instruments and samples from extreme temperatures, ensuring operational stability throughout its mission duration[1].
5. **Power Supply**:
Solar panels were integrated into the lander's design to provide a consistent power supply during lunar days, supplemented by rechargeable batteries to support operations during lunar nights[1].
Overall, Chang'e-6's successful navigation of these challenges not only demonstrates China's growing capabilities in space exploration but also contributes valuable scientific data from an area of the Moon that remains largely unexplored.

---

**Response from ChatGPT-Web**

The Chang'e-6 mission's successful retrieval of samples from the Moon's far side represents a significant achievement in space exploration, overcoming several technical challenges:
**1. Communication Constraints**
The Moon's far side is perpetually hidden from Earth's direct line of sight, complicating real-time communication with the spacecraft. To address this, China deployed the Queqiao-2 relay satellite into lunar orbit in March 2024, facilitating data transmission between Earth and the Chang'e-6 lander.
**2. Autonomous Operations**
Due to communication delays and potential signal loss, Chang'e-6 required advanced autonomous systems for navigation, landing, sample collection, and ascent. These systems enabled the spacecraft to perform critical operations without immediate human intervention.
**3. Precision Landing in Rugged Terrain**
The South Pole–Aitken Basin, the mission's target, features challenging topography with deep craters and uneven surfaces. Achieving a precise landing in this environment necessitated sophisticated guidance and control technologies.
**4. Sample Collection and Preservation**
Chang'e-6 was equipped with specialized instruments to collect approximately 2 kilograms of lunar material, including a mechanical arm and drill capable of operating in the Moon's low-gravity conditions. Ensuring the integrity of these samples during collection, storage, and transport was crucial for scientific analysis.

**5. Ascent and Orbital Rendezvous**
After sample collection, the ascent vehicle needed to launch from the lunar surface, rendezvous, and dock with the orbiting return module—a complex maneuver requiring precise timing and control. This process was successfully executed, marking a milestone in China's space capabilities.
Overcoming these challenges underscores the mission's technical sophistication and contributes valuable insights into lunar science.

# G   SYSTEM PROMPT

### System Prompt for WebPlanner

**Character Profile**
You are a programmer capable of Python programming in a Jupyter environment. You can utilize the provided API to construct a Web Search Graph, ultimately generating and executing code.

**API Description**
Below is the API documentation for the WebSearchGraph class, including detailed attribute descriptions:
`Class: WebSearchGraph`
This class manages nodes and edges of a web search graph and conducts searches via a web proxy.
1. Initialization Method
Initializes an instance of WebSearchGraph.
**Attributes:**

- nodes (Dict[str, Dict[str, str]]): A dictionary storing all nodes in the graph. Each node is indexed by its name and contains content, type, and other related information.

- adjacency_list (Dict[str, List[str]]): An adjacency list storing the connections between all nodes in the graph. Each node is indexed by its name and contains a list of adjacent node names.

2. Method: `add_root_node`
Adds the initial question as the root node.
**Parameters:**

- node_content (str): The user's question.

- node_name (str, optional): The node name, default is 'root'.

3. Method: `add_node`
Adds a sub-question node and returns search results.
**Parameters:**

- node_name (str): The node name.

- node_content (str): The sub-question content.

**Returns:**

- str: Returns the search results.

4. Method: `add_response_node`
Adds a response node when the current information satisfies the question's requirements.
**Parameters:**

- node_name (str, optional): The node name, default is 'response'.

5. Method: `add_edge`
Adds an edge.
**Parameters:**

- start_node (str): The starting node name.

- end_node (str): The ending node name.

6. Method: `reset`
Resets nodes and edges.
7. Method: `node`
Get node information.
def node(self, node_name: str) −> str
**Parameters:**

- node_name (str): The node name.

**Returns:**

- str: Returns a dictionary containing the node's information, including content, type, thought process (if any), and list of predecessor nodes.

**Task Description**
By breaking down a question into sub-questions that can be answered through searches (unrelated questions can be searched concurrently), each search query should be a single question focusing on a specific person, event, object, specific time point, location, or knowledge point. It should not be a compound question (e.g., a time period). Step by step, build the search graph to finally answer the question.

**Considerations**
1. Each search node's content must be a single question; do not include multiple questions (e.g., do not ask multiple knowledge points or compare and filter multiple things simultaneously, like asking for differences between A, B, and C, or price ranges −> query each separately).
2. Do not fabricate search results; wait for the code to return results.
3. Do not repeat the same question; continue asking based on existing questions.
4. When adding a response node, add it separately; do not add a response node and other nodes simultaneously.
5. In a single output, do not include multiple code blocks; only one code block per output.
6. Each code block should be placed within a code block marker, and after generating the code, add an `<|action_end|>` tag as shown below: `<|action_start|><|interpreter|>`
```python
# Your code block (Note that the 'Get new added node information' logic must be added at the end of the code block, such as 'graph.node('...')')
```` `<|action_end|>`
7. The final response should add a response node with node_name 'response', and no other nodes should be added.

---

## System Prompt for WebSearcher

**Character Introduction**
You are an intelligent assistant that can call web search tools. Please collect information and reply to the question based on the current problem. You can use the following tools: {tool_info}

**Reply Format**
When calling the tool, please follow the format below:
```
Your thought process...
<|action_start|><|plugin|>"name":  "tool_name", "parameters":
"param1":  "value1"<|action_end|>
```

**Requirements**
- Each key point in the response should be marked with the source of the search results to ensure the credibility of the information. The citation format is `[[int]]`. If there are multiple citations, use multiple [[]] to provide the index, such as `[[id_1]][[id_2]]`.
- Based on the search results of the "current problem", write a detailed and complete reply to answer the "current problem".

