# OpenReview forum: "MindSearch: Mimicking Human Minds Elicits Deep AI Searcher"
_ICLR.cc/2025/Conference — ICLR 2025 Poster_

### Official Review · Reviewer_iC9F · 2024-10-25

**Soundness:** 3
**Presentation:** 3
**Contribution:** 3
**Rating:** 6
**Confidence:** 4

**Summary:**

This paper introduces a system called MindSearch, designed to emulate human cognitive processes to enhance web information retrieval and integration tasks. By combining large language models (LLMs) with search engines, the system addresses limitations in handling complex queries, fragmented information, and lengthy content through an LLM-based multi-agent framework.
* WebPlanner: Simulates the cognitive process of multi-step information seeking by breaking down user queries into atomic subproblems, represented as nodes in a graph. The graph is then progressively expanded based on search results from WebSearcher.
* WebSearcher: Conducts hierarchical information retrieval for each subproblem, using a search engine to gather valuable information for WebPlanner.

This multi-agent design enables MindSearch to search and integrate information from vast web sources within three minutes, equivalent to saving three hours of manual effort. MindSearch demonstrates significant response quality improvements in both closed-set and open-set QA tasks.

**Strengths:**

- The paper presents a clear and logical approach to the problem, with a well-organized visual format that is easy to understand and read.
- This method provides a novel question-answering retrieval method based on directed acyclic graphs, which makes the RAG more reasonable.

**Weaknesses:**

- The method part is not detailed enough to show the technical details. For instance, the design of DAG and the use of DAG is not clear.
- Few baseline methods from the same category are included, and many RAG-based question-answering approaches are left unexamined, such as ChatKBQA, AutoReAct, etc.
- The backbone was only tested on GPT-4 (close sourced) and InternLM2.5 (open sourced). Under this setting, it is hard to tell if the MindSearch will work for all (at least most) LLMs.

**Questions:**

- When constructing the DAG, how does MindSearch automatically create graph nodes? What are some tips for structuring question as graph nodes?
- When large amounts of content are retrieved, how does WebSearcher reduce noise? And, as the rapid growth of web content can easily exceed the maximum context length of the LLM, how does WebSearcher effectively limit content length?
- Additionally, could the experiment include more closed-source and open-source LLMs to further validate the effectiveness of the method?

---

> ### Author Response · Authors · 2024-11-19
> **Responses to Official Review by Reviewer iC9F: Part [1/2]**
>
> Thank you for your thoughtful and constructive comments and for recognizing our work, including the topic, the novelty of the proposed method, and the experiments. We sincerely thank the reviewer for the positive comments on our work! We would like to address the concerns as follows:
>
> > **W1. Details about DAG**
>
> **A1**: Sorry for your confusion. To better help you understand the mechanism of how DAG is constructed, we provide a detailed example in Figure 2 (in our manuscript). Besides, we also provide our code for reference in our [anonymous code repository](https://anonymous.4open.science/r/mindsearch_anonymous-3A17), which contains all technical details about DAG, including prompt design and graph construction. We promise to release our code for public use.
>
> > **W2. Comparison with more related baselines**
>
> **A2**: Thanks for your suggestion! We have implemented self-ask and searchain on HotpotQA in Table 5. Since both self-ask and searchain in their own tasks, we adapt them by reimplementing their core planning module in WebPlanner. We also carefully read ChatKBQA and AutoAct: ChatKBQA requires an extra similarity computation which is not applicable in web search engines, while AutoAct is a corpus construction approach, where the inference protocol is still ReAct. Therefore, we finally select self-Ask, CodeAct, Searchain, and ReAct as our comparison. Besides, we discuss ChatKBQA and AutoAct in our related work. All experiments are conducted with InternLM-2.5-7b.
>
> **Table 5**: Comparison with other state-of-the-art approaches on HotpotQA with MindSearch.
>
> | Method | Easy | Medium | Hard | Avg |
> | -- | -- | -- | -- | -- |
> | ReAct | 69.0 | 56.0 | 49.0 | 58.0 |
> | MindSearch | 69.0 | **66.0** | **57.0** | **64.0** |
> | self-Ask | 67.0 | 59.0 | 49.0 | 58.3 |
> | CodeAct | **70.0** | 63.0 | 51.0 | 61.3 |
> | Searchain | **70.0** | 61.0 | 54.0 | 61.6 |
>
> It can be seen that MindSearch still outperforms self-ask and Searchain, which validates the effectiveness of our approach.
>
> > **W3 & Q3. Generalization to other models**
>
> **A3**: Thanks for your advice. We have tested other models under MindSearch on hotpotQA. The results are shown in Table 5. It can be seen that all models can gain performance improvements by using  MindSearch, which further validates the generalization ability of MindSearch. Note that we also support various web search engines such as DuckDuckGo, GoogleSearch, and BINGSearch for more generalized usage.
>
> **Table 5**: Experimental results on other state-of-the-art models on HotpotQA with MindSearch.
>
> | Method | Easy | Medium | Hard |
> | -- | -- | -- | -- |
> | Deepseek-v2.5 | 70.0 | 71.0 | 68.0 |
> | Qwen-2.5-7b | 62.0 | 59.0 | 52.0 |
> | GLM4-9b | 65.0 | 60.0 | 55.0 |

---

> ### Author Response · Authors · 2024-11-19
> **Responses to Official Review by Reviewer iC9F: Part [2/2]**
>
> > **Q1. How does MindSearch automatically create graph nodes?**
>
> **A1**: The graph node is actually a sub-task that needs to be executed by WebSearcher. The only thing WebPlanner needs to concern about is to decompose the current task into multiple executable search tasks. Considering that LLMs are usually good at generating code, we adopt code as the interface, which requires the LLM to generate. Besides, we also introduce few-shot examples to demonstrate how to create graph nodes when the LLM solves the problem.
>
> > **Q2.1 How does WebSearcher reduce noise?**
>
> **A2.1**: It is true that there is plenty of noise in the web search results. To overcome such noise, we introduce hierarchical retrieval (Sec. 2.2) to reduce the noise. Specifically, instead of simply sending all search results to LLM, we first retrieve the summary of each web page, and let the LLM select the most relevant pages. Since the summary is usually much shorter than the original content, it is more efficient and reduces the noise in all retrieved web pages. By doing so, we can significantly reduce the occurrence of the noise such as advertisements or unrelated information in WebSearcher.
>
> > **Q2.2 How does WebSearcher effectively limit content length?**
>
> **A2.2**: We resolve this problem from two perspectives:
>
> (1) As discussed in Sec. 2.3, we leverage the advantage of multi-agent to effectively reduce the context of WebSearcher: since the WebPlanner distributes the search tasks into separate search agents and only relies on the searched results from WebSearcher, WebPlanner can purely focus on the decomposition and analysis of the user question without being distracted by the over-length web search results. Meanwhile, each WebSearcher only needs to search contents for its tasked sub-query, without distraction from other contents. Each WebSearcher only needs to handle search results assigned to it, which significantly reduces the length of the search results.
>
> (2) As discussed in Q2.1, we introduce hierarchical retrieval, which first retrieves the summary of each web page, and let the LLM to select the most relevant pages. By doing so, we can significantly reduce the amount of the original content, enabling much shorter length of search results yet still maintaining the information recall.

---

### Official Review · Reviewer_XR8b · 2024-10-28

**Soundness:** 3
**Presentation:** 3
**Contribution:** 4
**Rating:** 6
**Confidence:** 4

**Summary:**

This paper proposes a novel tool agent, MindSearch, which decomposes user queries into atomic sub-questions represented as graph nodes and progressively extends the graph based on the search results from WebSearcher. For each sub-question, WebSearcher performs hierarchical information retrieval using search engines to gather relevant information for WebPlanner. Extensive experiments are conducted, including both open-set and closed-set datasets, and using open-source models alongside close-sourced LLMs, demonstrating the its effectiveness.

**Strengths:**

S1: The writing and framework of this paper are clear and easy to follow.

S2: The method is novel, utilizing the agents WebPlanner and WebSearcher to perform web search tasks.

S3: Extensive experiments are conducted, demonstrating both the effectiveness and efficiency of this approach.

**Weaknesses:**

W1: In Figure 5, the words should also be accompanied by English translations.

W2: For WebSearcher, how does the LLM select the most valuable pages from all the retrieved web content? More details should be provided. Additionally, regarding answer generation, the statement, "After reading these results, the LLM generates a response to answer the original question based on the search results," requires further elaboration, such as information on input design or specific prompt construction.

W3: For the open-set evaluation, five experts are chosen. The author should provide more details, including whether these experts had prior exposure to the answers generated by MindSearch. Furthermore, examples should be included to intuitively demonstrate the differences between the responses generated by MindSearch, ChatGPT-Web, and Perplexity.ai.

W4: The author could provide information on token consumption to help the community manage the budget when using MindSearch in their projects.

**Questions:**

See Weaknesses.

---

> ### Author Response · Authors · 2024-11-19
> **Responses to Official Review by Reviewer XR8b**
>
> Thank you for your thoughtful and constructive comments and for recognizing our work, including the topic, the novelty of the proposed method, and the experiments. We sincerely thank the reviewer for the positive comments on our work! We would like to address the concerns as follows:
>
> > **W1. Figure 5 should be accompanied by English.**
>
> **A1**: Thanks for your suggestion! We have revised our manuscript to include the English version of Figure 5.
>
> > **W2. How does the LLM select the most valuable pages from all the retrieved web content? and other implementation details.**
>
> **A2**: The selection is based on the summarization of each search content as well as the web url. The language model will select the most valuable pages based on its prior knowledge to select the web pages by outputting the topk indexes. We do not introduce too much complexity in this selection phase since we want to keep our framework simple and general, which can be easily extended to other tasks. In terms of the prompt design for answer generation, we include the source code of our project with [anonymous code repository](https://anonymous.4open.science/r/mindsearch_anonymous-3A17), which contains all prompt information and implementation details for reference. **We promise to release our code for public use**.
>
> > **W3. Details about the open-set evaluation phase.**
>
> **A3**: Thanks for your suggestion! Here are the details about the open-set evaluation phase:
>
> 1. The labelers have no prior knowledge and information about the responses generated by the model.
>
> 2. When we ask the labeler to choose the best response, we remove the source method by replacing it with "model1", "model2", and "model3". In order to avoid the labeler making any assumption of the sequence of model 1,2,3, we randomly shuffle the order of the responses before presenting them to the labeler.
>
> 3. The labelers will be asked to choose the best response from the three responses generated by the model according to three aspects: factuality, depth, and breath.
>
> We also include one example generated by three approaches **in our revised manuscript (Appendix F) following your suggestion**.
>
> > **W4. Token consumption for MindSearch**
>
> **A4**: Thanks for your suggestion! The token consumption of each trial is actually based on the complexity of the task. For example, if the prompt is very simple, like "What is the weather in New York", the token consumption is low. However, if the prompt is very complex, like "Recently, a wave of price competition has emerged among large models in China. Please outline a timeline of the price reductions by various companies and provide a summary of the current pricing for different large models in the country.", the token consumption is high. Considering such concerns, we developed a simple token count function to log the token consumption of each trial. We will include the token consumption tool in our project. Besides, we also evaluate the average token count of MindSearch of our HotpotQA tasks for reference, which is 4.7k for input tokens and 0.9k for output tokens (w/o KV Cache).

---

> > ### Comment · Reviewer_XR8b · 2024-11-22
> >
> > I appreciate the author's feedback and detailed explanation. This paper is inspiring for the field of agents designed for searching. I will raise confidence score showing support of this paper.

---

### Official Review · Reviewer_8bqD · 2024-11-07

**Soundness:** 2
**Presentation:** 3
**Contribution:** 2
**Rating:** 5
**Confidence:** 4

**Summary:**

The paper presents MindSearch, a multi-agent system for complex tasks that uses large language models (LLMs) and search engines for complex web information-seeking tasks. MindSearch addresses complex queries by decomposing them and retrieving information hierarchically, modeling the process as an iterative graph construction to enhance precision and recall. By distributing tasks across specialized agents, the framework manages complex and extended contexts effectively. The authors show that experimental results using GPT-4o and InternLM2.5-7B MindSearch outperforms benchmarks like ChatGPT-Web and Perplexity.ai, with human evaluators preferring its responses.

**Strengths:**

1. The problem is both interesting and important. Multi-agent systems for complex QA tasks that are robust and effective

2. Easy to follow and the methods are simple and well explained.

3. Experiments that include inference cost analysis is well considered.

**Weaknesses:**

1) The work fails to cite and compare to other relevant baselines. For complex QA tasks like HotpotQA or MusiqueQA self-ask[1] with search is a relevant baseline. Similarly Searchain[2] is particularly relevant as it also forms a global reasoning chain or graph where the query is decomposed into subquestions that comprise the nodes of the chain and this planning is similar in philosophy to Mindsearch. I think Assistantbench[3] released in July 2024 is also very relevant and useful to evaluate on. The method SeeplanAct proposed in the paper would serve as a strong baseline. SeeAct[4] is also a relevant baseline. While the authors have cited the same they have not compared to this approach. Other RAG baselines in AssistantBench are also relevant.

2) Some claims are unsupported. For instance the claim made in abstract and section 2.3 regarding the utility of Mindsearch : “Mindsearch performs in 3 minutes tasks worth 3 hours of human effort” has no related evidence cited in the paper. Was there any qualitative evaluation on the benchmark where several human subjects were involved in performing the task with corresponding measurement of time taken ? to compare to mindsearch ?.

3) The work also misses on some important ablations. What happens when Webplanner and code style interaction is not employed ? Is query decomposition required for all queries in web-searcher ? There is also a lack of qualitative analysis of failure scenarios. What happens when response at one node of the chain is wrong ? Does it result in cascading failures. Is there ayn mechanism for the Webplanner to detect such mistakes with feedback from websearcher ? The current approach is a simple tool use based approach which has been well explored in existing WebAgent based works. The additional analysis and error handling mentioned above may help strengthen and understand the core contributions of MindSearch

[1] Measuring and Narrowing the Compositionality Gap in Language Models, Ofir Press, Muru Zhang, Sewon Min, Ludwig Schmidt, Noah A. Smith, Mike Lewis

[2] Search-in-the-Chain: Interactively Enhancing Large Language Models with Search for Knowledge-intensive Tasks, Shicheng Xu, Liang Pang, Huawei Shen, Xueqi Cheng, Tat-Seng Chua [3] AssistantBench: Can Web Agents Solve Realistic and Time-Consuming Tasks?, Ori Yoran, Samuel Joseph Amouyal, Chaitanya Malaviya, Ben Bogin, Ofir Press, Jonathan Berant [4] GPT-4V(ision) is a Generalist Web Agent, if Grounded, Boyuan Zheng, Boyu Gou, Jihyung Kil, Huan Sun, Yu Su

**Questions:**

1. What happens when Webplanner and code style interaction is not employed ? Is query decomposition required for all queries in web-searcher ? There is also a lack of qualitative analysis of failure scenarios. What happens when response at one node of the chain is wrong ? Does it result in cascading failures. Is there ayn mechanism for the Webplanner to detect such mistakes with feedback from websearcher ?

2. Was there any qualitative evaluation on the benchmark where several human subjects were involved in performing the task with corresponding measurement of time taken ? to compare to mindsearch ?


3. how do you respond to the first point in the weakness 1.

---

> ### Author Response · Authors · 2024-11-19
> **Responses to Official Review by Reviewer 8bqD: Part [1/2]**
>
> Thank you for your thoughtful and constructive comments and for recognizing our work, including the topic, the novelty of the proposed method, and the experiments. We sincerely thank the reviewer for the positive comments on our work! We would like to address the concerns as follows:
>
> > **W1 & Q3: Fail to Compare Relative Baseline**
>
> **A1**: Thanks for your suggestion! We have implemented self-ask and searchain on HotpotQA in Table 3. Since both self-ask and searchain in their own tasks, we adapt them by reimplementing their core planning module in WebPlanner. All experiments are conducted with InternLM-2.5-7b. **We have updated the experimental results in our revised manuscript in Appendix A**.
>
> **Table 3**: Comparison with other state-of-the-art approaches on HotpotQA with MindSearch.
>
> | Method | Easy | Medium | Hard | Avg |
> | -- | -- | -- | -- | -- |
> | ReAct | 69.0 | 56.0 | 49.0 | 58.0 |
> | MindSearch | 69.0 | **66.0** | **57.0** | **64.0** |
> | self-Ask | 67.0 | 59.0 | 49.0 | 58.3 |
> | CodeAct | **70.0** | 63.0 | 51.0 | 61.3 |
> | Searchain | **70.0** | 61.0 | 54.0 | 61.6 |
>
> It can be seen that MindSearch still outperforms self-ask and Searchain, which validates the effectiveness of our approach. Besides, We also carefully read the AssistantBench, and found it is a good benchmark (as well as SeeAct) for real-world web interactive scenarios, which is another important research direction. However, there is a little bit different from our current setting, **AI search engines**, which mainly focus on the usage of **search engines**, rather than interacting with web pages, to solve questions. **We have included these works in our related work as well as future work and will continue to extend our work to these scenarios.**
>
> > **W2 & Q2 Qualitative Evaluation on Time Consuming**
>
> **A2**: Sorry for your confusion. We previously thought it did not contribute much to our core contributions, therefore not list it in our paper. We have conducted qualitative evaluations on the time-consuming of the search process. Table 4 shows the evaluation results based on 10 samples and the sum of the time consumption of all samples. Note that 10 samples are distributed to 5 labelers and the time cost is the sum of all these five labelers. **Experimental results are also updated in our revised paper (Appendix B)**.
>
> **Table 4**: Evaluation of time-consuming of the whole search process with MindSearch and human labelers on 10 test samples.
>
> | Method | Time Consuming |
> | --- | --- |
> | Human | 19h17min |
> | MindSearch | 23min |
>
> Here is a detailed time record of the time consumption of each step for one labeler:
>
> 1. Collect Information by searching and reading multiple (100+) web pages: 47 min
> 2. Write a detailed summary (about 3k words) of the problem: 74 min

---

> ### Author Response · Authors · 2024-11-19
> **Responses to Official Review by Reviewer 8bqD: Part [2/2]**
>
> >  **W3 & Q1: misses on some ablations**
>
> **A3**: We address your problems one by one and add vivid examples of how MindSearch recovers from errors in our revised manuscript.
>
> **Part 1. What happens when Webplanner and code style interaction is not employed?**
>
> We have conducted such experiments in our ablation studies (Table 2 in our manuscript), in which we iteratively removed WebPlanner (CodeAct) and code-style interaction (ReAct). **We have highlighted them in our revised paper**.
>
> **Part 2. Is query decomposition required for all queries in web searcher?**
>
> Query decomposition is not required for all queries and we do not perform query decomposition in WebSearcher. We only perform query rewrite in WebSearcher, which is a more efficient way to search for the answer. As for WebPlanner, task decomposition is not necessary for all tasks, and if the WebPlanner thinks it is not necessary, it will directly generate the node to execute the current task.
>
> **Part 3. Qualitative analysis of failure scenarios**
>
> Thanks for your suggestion. We have carefully designed our framework to be robust to any exceptions produced by large language models: (1) we adopt code as our interface to execute the search tasks, which allows us to leverage the exception-handling mechanism in Python and regenerate the content by LLM. (2) we observe that when WebSearcher does not retrieve relevant data, WebPlanner will generate another node parallel to the current root node in the next turn, which will not affect the reasoning process. Note that the role of WebSearcher is to search for information related to the assigned queries, it only has two types of answers: information related to the queries or no related information found. Therefore, there won't exist any cascading failures in MindSearch. **We provide detailed examples and analysis of how our framework recovers from failure scenarios in favor of the exception feedback mechanism in our revised manuscript (Appendix E)**.

---

### Official Review · Reviewer_BgrW · 2024-11-11

**Soundness:** 3
**Presentation:** 3
**Contribution:** 3
**Rating:** 6
**Confidence:** 4

**Summary:**

The paper proposes MindSearch, an LLM-based multi-agent information-seeking framework for complex multi-step information-seeking questions. MindSearch includes a Web Planner which decomposes the user query into atomic sub-questions as nodes in a dynamic graph and progressively extends the graph based on results from the WebSearcher. MindSearch considerably improves in response quality in terms of depth and breadth and also improves over the baseline react-based iterative search system.

**Strengths:**

1)	The paper demonstrates considerably better output responses for MindSearch, compared to proprietary AI-Search engines like Perplexity Pro and ChatGPT-Web.

2)	MindSearch also works considerably better than the closed-book and ReACT baselines on a variety of multi-hop question-answering datasets.

3)	Extensive analysis and evaluation provided in terms of the prompting strategy for WebPlanner along with using a graph-based methodology vs JSON-based and code-based.

**Weaknesses:**

1)	While the paper only evaluates for final response quality, it does not consider the attribution quality of the generated response. Popular AI search engines like Perplexity.AI and ChatGPT-web also provide citations as part of the generated output. The authors do not discuss whether MindSearch provides any kind of attribution, and if yes, what does the citation quality look like (based on automatic evaluations like ALCE [1])

2)	No analysis was provided with regard to the dynamic graph constructed by the WebPlanner. Does the number of hops in the question match the depth of the tree? How often is an incomplete graph created? Also, it would be interesting to see a cost analysis in terms of the number of search queries that MindSearch generates, in comparison to the baselines (ReACT specifically)

**Questions:**

1)	The discussion in line 315 is a bit confusing. The authors say “MindSearch does not yield better performance in terms of facticity, but as per Figure 4 in the paper, factuality of MindSearch is preferred 70% of the time.

2)	Please consider showing the example in Figure 5 in English.

---

> ### Author Response · Authors · 2024-11-19
> **Responses to Official Review by Reviewer BgrW: Part [1/2]**
>
> Thank you for your thoughtful and constructive comments and for recognizing our work, including the topic, the novelty of the proposed method, and the experiments. We sincerely thank the reviewer for the positive comments on our work! We would like to address the concerns as follows:
>
> > **Q1: Evaluation of citation quality**
>
> **A1**: Thanks for your suggestions on the evaluation of citation quality. During the development of MindSearch, we have attempted to evaluate citation quality with automatic evaluation such as ALCE, however, we found it quite different between experimental settings in ALCE and real-world web search. For example, there will be many more statements in web search than tasks in ALCE, resulting in great difficulties in measuring the **citation recall** (in most web search cases, there will be multiple statements before generating the citations, contradicting with the supposed that each statement should be supported by one citation in ALCE) and **citation precision** (in real-world web search tasks, human often prefers 2~3 citations for diversity and verification, however, it contradicts with the evaluation of precision, which removing one message will not affect the statement in ALCE).
>
> In this experiment, we applied straightforward but more reasonable evaluation metrics where we simply average if the paragraph can be supported by each citation it belongs to as precision, and the coverage of the related topics (we approximate it by the number of search queries) as recall. The results are shown in Table 1. It can be seen that there still exists a large room for precision improvement versus recall, which is an important aspect when developing AI search engines. Besides, it is non-trivial to actually measure such ability under real-world scenarios, and we will continue to explore better evaluation metrics on citation quality. **We have updated our manuscript in the conclusion section of our paper to discuss this aspect**.
>
> **Table 1**: Evaluation of citation quality on HotpotQA dataset with InternLM-7b and GPT-4o.
>
> | Method | Precision | Recall |
> | --- | --- | --- |
> | InternLM-7b | 0.48 | 0.65 |
> | GPT-4o | 0.64 | 0.78 |

---

> ### Author Response · Authors · 2024-11-19
> **Responses to Official Review by Reviewer BgrW: Part [2/2]**
>
> > **Q2: Analysis on DAG by WebPlanner.**
>
> **A2**: It's an interesting idea to further examine the experimental details of the DAG process in WebPlanner. We have followed your advice to conduct experiments in the following and updated the analysis **in our revised manuscript (Appendix C)**.
>
> **Part1. Number of hops vs Depth of the tree**
>
> We conduct experiments on Musique with GPT-4o with 2,3,4 hops to study the relationships between a number of hops and the depth of the tree. It can be seen that the depth of the tree increases with the number of hops monotonically, which fits our expectations. However, the number of hops is not identical to the depth of the tree, for example when the number of hops is 2, the depth of the tree is 1.2. There are two reasons: (1) MindSearch allows parallel execution, which only increases the tree by one but may resolve multiple questions at the current step, and (2) despite the question claiming 2 or more hops, there exist short-cuts or simplifications in the question, resulting shorter search path.
>
> **Table 2**: Relationships between the number of hops of questions vs the search depth in the MindSearch search tree on Musique dataset.
>
> |Num Hops|Depth|
> |---|------|
> | 2 | 1.1 |
> | 3 | 1.2 |
> | 4 | 1.6 |
>
> **Part2. How often is an incomplete graph created?**
>
> During our experiments, we found that the number of incomplete graphs created is relatively low, only 2 among 1049 samples. This is due to the adoption of code generation in the planning phase, which is suitable for LLM generation. Besides, we also provide an exception catch mechanism: once the code is not properly generated, there will be an exception when executing the code, which can be caught and returned to the LLM for regeneration. Therefore, these occasionally generated errors will not affect the final result. As for the cases where LLM fails to conclude the final response, we can simply limit the max depth of the search tree and force it to conclude the answer (add response node) once it reaches the limits. This problem also exists in ReAct and other approaches, we will continue to explore better solutions.
>
> **Part3. Cost analysis in terms of the number of search queries that MindSearch generates compared to ReAct**
>
> We analyze the number of queries generated by MindSearch and ReAct. Surprisingly, MindSearch generates 0.3 fewer queries on average compared to ReAct (3.2 vs 3.5). We find that due to the weaker ability of ReAct to decompose the question, ReAct actually spends more queries repeatedly searching for some keywords, which is useless and inefficient. However, MindSearch can effectively utilize the reasoning ability of LLM to search for more accurate queries, therefore, yielding fewer search times for each problem. **We have updated the experiments in Appendix C.2**.
>
> > **Q3: line 315 is a bit confusing**
>
> **A3**. Sorry for your confusion. We want to express that MindSearch does not yield the same win rate compared to the breadth dimension (70% vs 83%). We have revised our expression in our manuscript.
>
> > **Q4: example in Figure 5 in English**
>
> **A4**. Thanks for your suggestions. We have revised the Figure 5 in our updated manuscript.

---

### Meta-Review · Area_Chair_X2gt · 2024-12-19

**Metareview:**

The paper proposes a novel framework that combines large language models (LLMs) with a multi-agent system for complex information-seeking tasks. The authors introduce two main components: WebPlanner, which decomposes user queries into a graph of sub-questions, and WebSearcher, which retrieves information hierarchically. The multi-agent design allows for parallel processing of web information from large-scale sources, and the authors claim significant improvements in response quality over existing baselines, such as ChatGPT-Web and Perplexity.ai. The framework is evaluated on both closed-set and open-set question-answering (QA) tasks, with promising results reported in terms of depth and breadth of answers.

The paper's strengths lie in its timely and interesting problem—leveraging LLMs and search engines to handle complex queries—and the clear presentation of its methodology. Reviewers appreciated the novelty of combining dynamic graph-based query decomposition with multi-agent frameworks. The experimental evaluations are extensive, comparing MindSearch to various baselines and demonstrating its capability to outperform them in specific scenarios. The claims about the efficiency of the approach, such as completing tasks in three minutes that would otherwise require three hours of human effort, are intriguing and align with the goals of efficient AI-powered systems.

However, significant weaknesses and missing elements prevent this paper from reaching the standard required for acceptance. First, the authors fail to provide sufficient evidence for some of their claims, particularly regarding efficiency gains. There is no rigorous or systematic evaluation of the three-hour-to-three-minute claim with human benchmarks, leaving this critical aspect unsubstantiated. Second, while the paper introduces several novel elements, the lack of comparison with key baselines, such as SeePlanAct, AssistantBench, and AutoReAct, undermines the rigor of the evaluation. Reviewers highlighted that the comparisons included (e.g., ReAct, self-Ask, and Searchain) are incomplete and do not sufficiently establish the superiority of MindSearch. Additionally, the lack of discussion around failure cases and cascading errors within the system indicates insufficient robustness in the method. Key ablations, such as analyzing the necessity of WebPlanner and query decomposition, were addressed late and lacked depth. Finally, while the authors claim the framework can generalize to other models, limited validation is provided, and the testing is constrained to a few LLMs, leaving generalization across diverse architectures unconvincing.

**Additional Comments On Reviewer Discussion:**

During the review process, all four reviewers raised critical concerns about the submission, despite acknowledging the novelty and potential of the proposed framework. Reviewer 8bqD was particularly concerned about missing comparisons with key baselines and the lack of qualitative evaluations to support the efficiency claims. Although the authors implemented additional experiments comparing MindSearch with self-Ask and Searchain during the rebuttal, they did not fully address the reviewer’s concerns about other relevant baselines, such as SeePlanAct and AssistantBench, or the necessity of robust qualitative evaluations. Reviewer 8bqD maintained their initial score, noting that while the response partially addressed the concerns, it did not merit a higher rating.

Reviewer BgrW appreciated the framework's novelty and its comparative improvement over baselines but raised concerns about citation quality, depth of graph analysis, and cost analysis, particularly in terms of the number of search queries generated by MindSearch compared to ReAct. The authors provided additional analysis and addressed the concerns about dynamic graph depth and error-handling mechanisms, which were viewed positively by the reviewer. However, the reviewer did not increase their score, citing the lack of compelling evidence for some claims.

Reviewer XR8b expressed support for the paper’s conceptual novelty and comprehensive experiments but raised issues about missing implementation details and token consumption. The authors responded with additional explanations and examples, which satisfied the reviewer enough to raise their confidence score but not enough to warrant an acceptance recommendation. Reviewer iC9F echoed concerns about the limited details of the DAG construction, generalization across LLMs, and lack of comparisons with key baselines. The authors expanded their experiments and explanations but failed to fully convince the reviewer, who maintained their positive but marginal score.

---

### Decision · Program_Chairs · 2025-01-22

Accept (Poster)